# Development of a perioperative thermal insulation system: Testing comfort properties for different textile sets

Isaura Carvalho[1,2], Miguel Carvalho[3], Liliana Fontes[3], Teresa Martins[2,4]*, Fernando Abelha[5,6]

1 Department of Operating Room, Hospital da Prelada, Porto, Portugal, 2 Center for Health Technology and Services Research (CINTESIS@RISE) Porto, Porto, Portugal, 3 Department of Textile Engineering, University of Minho, Guimarães, Portugal, 4 Escola Superior de Enfermagem do Porto, Porto, Portugal, 5 Medical Faculty University of Porto, Porto, Portugal, 6 Director of the Anesthesiology Service, Centro Hospitalar Universitário S. João, Porto, Portugal

☯ These authors contributed equally to this work.
* teresam@esenf.pt

**Data Availability Statement:** All relevant data are within the manuscript. Supplementary data files are

## Abstract

The poorly physical and psychological conditions of the patients make the body thermal protection crucial in the perioperative context, due to the risk of hypothermia. The lack of evidence regarding the effectiveness of textile coverings in protecting patients in the operating room, underscores the recommendation of the forced warming system using non-woven fabric for ensuring the best thermal protection in the perioperative context. This study is part of a development process of a three-layered thermal insulation system, a blanket for use in the perioperative context. After previous selection of two fabrics for the mid and outer layers, in this study three fabric samples for the inner layer with same soft tactile sensation and different textile compositions were tested to find its effect on increasing the thermal insulation of the whole set, using a thermal manikin. The serial method was used to calculate the thermal insulation properties of the sets. The best thermal insulation and thermal comfort performance was obtained by the set using an inner layer composed of polypropylene, polyamide, and elastane whose results were the highest thermal conductivity and thickness and the lowest maximum stationary heat flow density. The results indicated that this fabric influenced positively the values of the whole set once increased its thermal protection effectiveness when compared to the other tested sets. This set is more suitable for future testing in patients during their stay in the perioperative setting.

## Introduction

Human life depends on the body's ability to maintain its internal temperature at around 37˚C. Due to their homeothermic ability, humans can protect themselves from thermal environmental hostility. Their internal mechanisms, which involve complex biochemical reactions and behaviours that stimulate self-protection actions, are activated by cold or heat and give them thermal sensations and comfort responses [1]. In specific situations of extreme thermal

available: https://doi.org/10.5061/dryad.
np5hqbzx7.

**Funding:** This work is financed by national funds
through FCT - Portuguese Foundation for Science
and Technology under the project UIDB/4255/2020
and reference UIDP/4255/2020. The funders had
no role in study design, data collection, and
analysis, decision to publish, or preparation of the
manuscript.

**Competing interests:** The authors have declared
that no competing interests exist.

adversity or body fragility, such as a person with a disease condition in the surgical context,
those mechanisms are impaired, and the person is at risk of hypothermia. Inadvertent periop-
erative hypothermia occurs in about 50 to 90% of patients undergoing surgery lasting longer
than 60 minutes [2–5].

Significant perioperative hypothermia-related complications were found to cause an
increase in surgical wound infection by up to 300%, blood loss by 16%, cardiovascular events
by up to 50%, and hospital stay by up to 20% [6–11]. Moreover, the costs of post-operative
infection in a patient undergoing total hip arthroplasty can sum up to 40.000 euros [12, 13].
Before surgery, the patient's condition is poor due to biochemical changes triggered by health
problems, anxiety, fasting, inadequate gowns (often of thin fabric), the effect of anaesthetic
drugs, low room temperature, and air conditioning. These factors can increase the tempera-
ture drop and the thermal discomfort sensation and impair the recovery process [6–11], thus
highlighting the need for suitable patient thermal protection.

Two types of body protection can be used in the perioperative context: passive measures or
thermal insulation systems, which create a barrier to prevent heat loss from the patient's body
to the environment, such as cotton sheets or cotton blankets, and active measures or warming
systems, which warm the patient's body through heat from external sources, such as water cir-
culating mattresses and forced air warming blankets. The lack of evidence either on the effec-
tiveness of existing thermal insulation systems (passive measures) or on the development of
new ones, make the warming systems the recommended active measures for patient protection
in the perioperative context. In a study carried out with women undergoing caesarean, in the
intraoperative phase, the application of a warm air system in the intervention group and 3 pre-
heated cotton sheets in the control group were compared. It was found that, after 30 minutes,
the control group reported lower thermal comfort [14].

Contrary to thermal insulation systems, several warming systems have been well developed
and evidence shows their greater effectiveness when compared to passive measures. Two stud-
ies have been carried out to understand their effects on thermal protection and inpatient com-
fort. In both studies, active measures (forced air warming systems and water circulating water
mattresses) proved to be more effective in controlling perioperative hypothermia than passive
measures (cotton sheets) [15, 16].

The most widely active measure adopted is a forced air warming system, consisting of an
electrical unit, a tubular sleeve, and a disposable non-woven blanket with void spaces inside to
fill with warm air. The warmed air is driven through the sleeve from the electrical unit to the
blanket placed on the patient to facilitate body warming [17].

However, in daily practice in the operating room, health professionals are often aware of the
forced air warming blanket-related disadvantages, such as (1) the frequent discomfort of
patients and permanent discomfort of the surgical teams due to the emission of heat, (2) the
limitations of physical space needed for the equipment, (3) the energy dependence, (4) the need
for maintenance and the possibility of breakdown, (5) the production of waste caused by single-
use consumables, and (6) an additional source of noise. Some authors also refer to the high pos-
sibility of contamination of the warmed air transport sleeve from the engine to the blanket since
its rough configuration makes adequate cleaning difficult [18]. Furthermore, the structural
weaknesses of thermal insulation systems often make them insufficient to keep patients warm.

In an attempt to develop a passive system that could be more effective than the existing
ones, the authors were inspired by the recommended layered clothing composed of three sepa-
rate garments that is an effective thermal protection system that keeps people warm and dry
while performing activities in adverse conditions, such as mountaineering and sailing. Then, a
blanket made with three different overlapping fabrics, in which fabric corresponds to a layer,
was designed [19].

The blanket is a thermal insulation system because it creates a barrier to prevent heat loss from the patient's body to the environment. Although the fabrics used in this study already exist on the textile market and are used in some types of clothing, no evidence of their use in the perioperative context was found.

This blanket was intended to be used in the operating room to keep patients warm during the intraoperative period. Since the criteria for the mid and outer layers were the high permeability to air for the mid layer and the waterproof for the outer layer, two fabrics had been previously chosen and tested.

Therefore, this study aimed to analyse the thermal properties of three different fabrics with similar tactile sensations proposed for the inner layer, and to find which fabric contributes most significantly to increase the thermal protection of the whole three-layered blanket, using a thermal manikin.

## Materials and methods

The three-layered blanket concept has never been tested in the perioperative context.

Despite this new blanket is inspired by the three-layered model used in clothing for extreme activities, some differences were considered in the characteristics of the fabrics to be chosen. Therefore, the following criteria were considered: (1) the blanket would be composed of existing and usual fabrics, since developing new textile materials and fabrics is not an objective of this research; (2) development of a blanket suitable for patients in the lying position; (3) the use a soft tactile fabric for the inner layer, a low-density and highly porous fabric for the mid-layer, and an impervious air fabric for the outer layer; (4) although the fabrics used in extreme activities have technical properties such as breathability and perspiration, those characteristics would not be considered as important for this new solution since the humidity of the sweat produced by the person during surgery is incipient; (5) development of a solution capable of mechanical washing and disinfection; (6) the ease of penetration of hot air inside, its return to the patient and the reduction of heat escape to the environment.

In this blanket, fabrics are layered and stitched around the edges. Heat emitted from the patient enters the interior of the blanket through the fibers of the inner layer. Then the heat circulates within the fibers of the mid layer. The outer layer prevents heat from escaping to the outside and prevents cold from entering the interior of the blanket. By remaining within the blanket, the heat keeps the patient warm and comfortable.

The overlay of the blanket´s fabrics is schematically depicted in Fig 1.

The selection of the fabrics for the construction of the innovative blanket was guided by the objectives associated with each layer. These fabrics are accessible within the textile market and found application in various clothing types. The process of fabric selection was executed in collaboration with manufacturers.

For the mid layer, a polyester fabric with high openness and volume, lightweight and low-density was chosen. This fabric facilitates the transfer of heat generated by the body through the spaces between its fibers, allowing the accumulation of a cushion of warm air, and ensuring a sensation of comfort without increasing significant weight or volume [19–21].

The chosen outer layer consists of a waterproof and slightly heavier fabric, crafted from polyester and polyurethane, designed to retain heat within the blanket and to prevent heat from escaping into the environment.

The air permeability of both fabrics was tested.

When considering the intended air permeability of the fabrics [22, 23], the evaluation of the outer and mid layer materials was conducted in accordance with NP EN ISO 9237 standards.

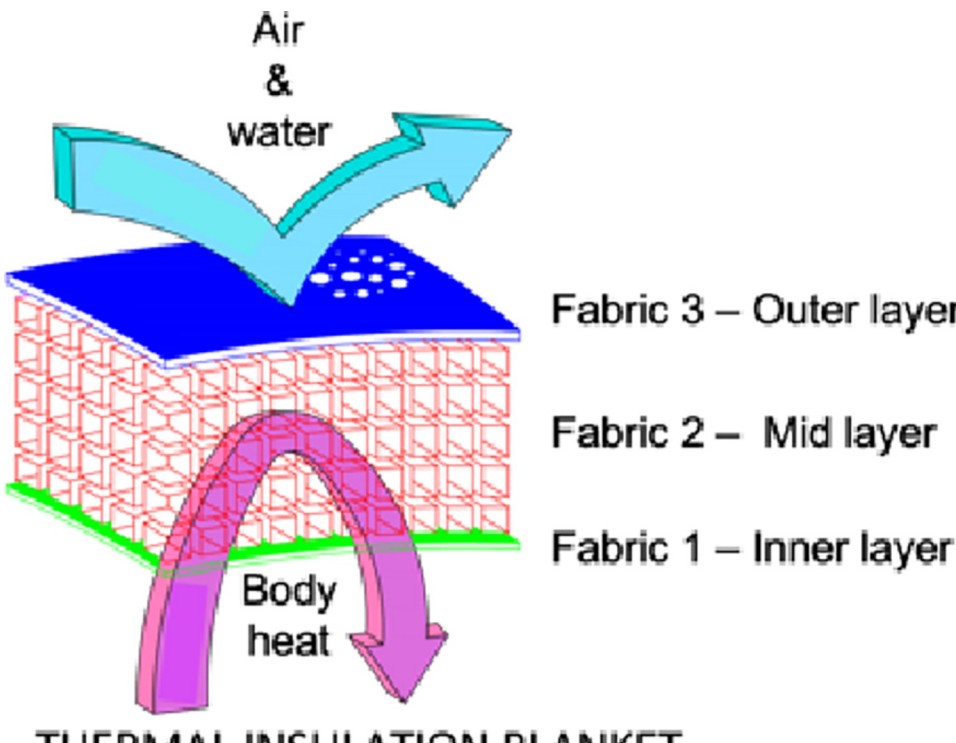

**Fig 1. The overlay of fabrics of the thermal insulation blanket.**

The Textest FX Air Permeability Tester equipment was used, with a pressure of 100 Pa and a test surface of 20 cm$^2$.

The results indicated that the fabric chosen for the mid layer exhibited a high air permeability of 500 l/m$^2$/s, while the fabric selected for the outer layer demonstrated an air imperviousness of 0.3 l/m$^2$/s.

The inner layer, also known as the base layer or second skin due to its direct contact with the body, promotes comfort through (1) tactile softness, (2) elimination of condensation and preventing both skin and fabric surface from becoming wet and (3) the ability for quick fabric drying [19, 20, 24]. Maintaining dryness and warmth for the skin and the inner layer is crucial, as damp clothing allows heat to escape at a rate twenty-five times faster than dry clothing, leading to significant discomfort [19, 20].

According to experts in high mountain activities, inner layer fabrics in dark colours absorb more heat, thus keeping the body warmer and drying faster while lighter colours absorb less heat, making them more suitable for hot days [20]. Inner layer fabrics can be manufactured from various types of fibers. There seems to be an advantage in using synthetic fibers over natural ones, as they offer better thermal insulation and a greater comfort sensation, particularly when wet. Selecting the appropriate inner layer depends on the conditions and the type of activity performed, ensuring maximum comfort for the wearer [19]. Based on this information, to choose a synthetic and tactile softness fabric was the main criteria. During the selection process of fabrics for the inner layer, three fabrics with these characteristics were found. Two of the fabrics included polyester in their composition and were brown in colour. The other fabric included polyamide in its composition and was black in colour.

Despite the differences in their compositions, all three fabrics shared a similar tactile sensation, which led to the decision to test each of them.

**Table 1. Composition, structure, and weight of the fabrics.**

| Sample | Structure | Composition | Weight (g/m$^2$) |
|---|---|---|---|
| Outer Layer (A) | Warp Knitting with PU coating | 80% Polyester / 20% polyurethane | 300.07 |
| Mid Layer (B) | Non-woven | 100% Polyester | 35.28 |
| Inner Layer 1 (T1) | Knitting–Jersey | 61% Polypropylene / 34% Polyamide / 5% Elastane | 217.66 |
| Inner Layer 2 (T2) | Knitting–Jersey | 46% Polypropylene / 51% Polyester / 3% Elastane | 154.53 |
| Inner Layer 3 (T3) | Knitting–Jersey | 36% Polypropylene / 59% Polyester / 5% Elastane | 178.48 |

The composition, structure, and weight of the fabrics for the outer and mid layers and the three proposals for the inner layer are summarized in Table 1.

Considering the relevance of the fabrics for the inner layer since it is the one in close contact with the patient´s body, and considering the differences in their composition, the thermal comfort properties of these fabrics were tested for sensorial comfort.

The following thermal properties of the three samples proposed for the inner layer were tested: thermal conductivity, which measures the ability of a material to allow the flow of heat from the hottest to the coldest surface and determined by the thermal energy transferred per unit of time and per unit of surface, divided by the temperature gradient [23]; thermal diffusivity, the ability of heat to flow through the air in the fabric structure, and known as the transient thermal characteristic of textiles [25, 26]; thermal absorptivity, which is the surface property that allows the assessment of the fabric's character in relation to its 'warm/pleasant' feeling or instant heat flow that occurs when two bodies with different temperatures come into contact [25, 26]; thermal resistance, the characteristic that indicates the fabric's ability to resist the heat flow passing through it, which depends on the conductivity thickness of the materials and the temperature difference as an external factor [25, 26]; thickness, which refers to the perpendicular distance of the fabric, determining the dimension between its upper and lower sides [27]; and the maximum density of the stationary heat flux, which corresponds to the maximum value of heat exchanged between tissue and human skin during contact [21, 24]. The tests were made using the Alambeta apparatus, Model SENSORA, developed by Liberec Register Company from the Czech Republic, which objectively assesses the hot/cold sensation, simulating the heat flow between the human skin and the textile material. This sensation is an important indicator, not only when one touches a fabric but during frequent contact of the inner parts of the blanket with the skin [22].

Since three samples were proposed for the inner layer, three-layered sets with the same mid and outer layers were developed, overlapping one fabric over the other, to test how the inner layer affected the thermal protection performance of the whole set. Thermal resistance and thermal insulation of the three-layered sets were tested using a thermal manikin. Thermal manikins allow for the rigorous and safe thermal testing of clothing and an increased understanding of the thermal properties of different tested materials. With the configuration of an adult human body divided into thermally independent segments, these models can simulate and reproduce some physiological interaction processes between the human body and the environment in a laboratory setting with accuracy [28, 29]. Due to their characteristics, thermal manikins allow researchers to study, for example, the body temperature characteristics in neutral or cold conditions at low activity levels [28].

The thermal manikin used in this research was manufactured by PT.Technic from Denmark, 2009, with the body divided into 20 independent thermal segments, where the dry heat transfer takes place in one direction, from the inside of the manikin to the environment. This electric model is heated throughout its surface to achieve a constant temperature. This temperature can be adjusted to the desired values and ensure a balanced distribution across its

**Table 2. The body segments of the thermal manikin.**

| No. | Group A | Group B |
|---|---|---|
| 1 | L. Foot | Face |
| 2 | R. Foot | Skull |
| 3 | L. Low leg | L. Hand |
| 4 | R. Low leg | R. Hand |
| 5 | L. Front thigh | L. Forearm |
| 6 | R. Front thigh | R. Forearm |
| 7 | L. Back thigh | L. Upper arm |
| 8 | R. Back thigh | R. Upper arm |
| 9 | Pelvis | Chest |
| 10 | Back side | Back |

surface, like the human body. The power required to maintain a constant temperature is measured and correlated with thermal comfort [28]. The body segments of the thermal manikin are divided into two different groups for easy measurement and determination of thermal changings in the body segments. The body segments of each group are exhibited in Table 2.

Tests were conducted in an adiabatic chamber under controlled environmental conditions, according to ISO 15831, in the Laboratories of the Textile Engineering Department at the University of Minho, Portugal. The test time for measuring each three-layered set was 20 minutes, totalling 60 measurements. The sets were placed on top of the manikin´s upper body, simulating a patient undergoing surgery in the lower body. All tests were carried out once the temperature of all manikin segments stabilised at 33˚C (±0.001). The average room temperature was 22˚C (±0.5), the relative humidity was 42% (±2), and the air velocity was below 0.15 m/s, monitored by computer. These parameters were monitored continuously throughout the test. Results were automatically recorded and stored at the end of each evaluation. The manikin was placed lying on a bed in a static position, mimicking the position of a patient on the operating table, as illustrated in Fig 2.

## Thermal insulation calculation models

The thermal insulation of the tested three-layered sets can be calculated in two ways: by adding the area-weighted local thermal insulation to the different body segments of the manikin—serial method—or by using the heat flow from the manikin´s body-parallel method [23, 29].

In the Serial Model, which corresponds to the surface area-weighted thermal insulation, the total thermal insulation, $I_t$, or the resultant total thermal insulation, $I_{tr}$, are calculated from the test results obtained with the manikin, either stationary or moving its legs and arms, using the equation:

$$I_t \ or \ I_{tr} = \sum_i f_i \ x \left[ \frac{(T_{si} - T_a) \ x \ a_i}{H_{ci}} \right] \ \left( ^\circ Km2/W \right)$$

where

$$f_i = \frac{a_i}{A}$$

In the Parallel model, related to the surface area-averaged thermal insulation, the total thermal insulation, $I_t$, or the resultant total thermal insulation, $I_{tr}$, is calculated on the test results obtained with the manikin either stationary or moving its legs and arms, respectively, using

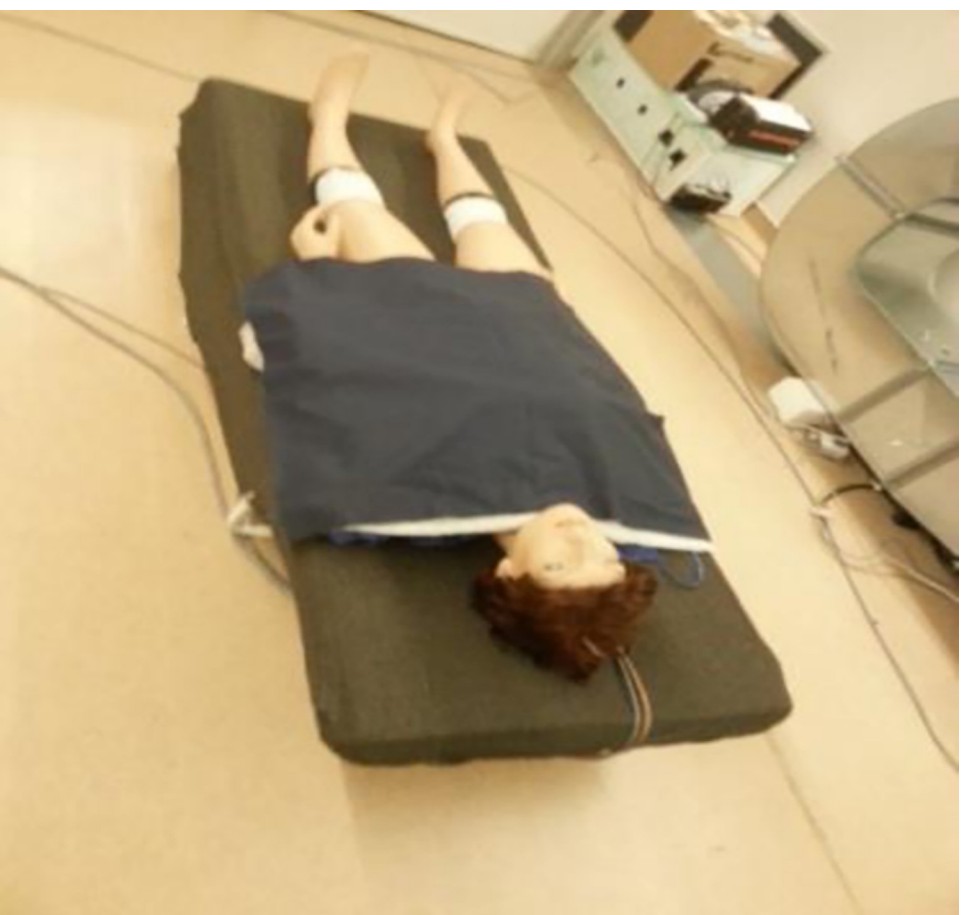

**Fig 2. The thermal manikin position.**

the equation:

$$I_t \text{ or } I_{tr} = \left[\frac{(T_{si} - T_a) \, x \, A}{H_c}\right] \left(^{\circ}Km2/W\right)$$

where

$$T_s = \sum_i f_i \, x \, T_{si} \, (^{\circ}\text{C})$$

$$H_c = \sum_i H_{ci} \, (W)$$

$I_t$—total thermal insulation of the clothing ensemble with the manikin stationary, in square meter kelvins per watt;

$T_{si}$—local surface temperature of section i of the manikin, in degrees Celsius;

$T_a$—air temperature in degrees Celsius;

$a_i$—surface area of section i of the manikin, in square meters;

$H_{ci}$—local heat loss from section i of the manikin, in watts;

$A$—total body surface area of the nude manikin, in square meters;

$H_c$—heat loss from the total surface area of the manikin´s body;

Table 3. The values for inner layers, calculated by Alambeta apparatus.

| SAMPLE TYPE | VARIABLE VALUES (MEAN AND SD) | | | | | |
|---|---|---|---|---|---|---|
| | $\lambda$ W/(m.k) | $\alpha$ (mm²/s) | $\beta$ (nm) | $\tau$ (m²/kw) | $h$ (m) | $q$ max (W/m²) |
| T1 | 57.76 (1.06) | 0.1364 (0.014) | 156.8 (5.71) | 6.54 (0.32) | 0.956 (0.013) | 0.812 (0.023) |
| T2 | 47.56 (1.49) | 0.1982 (0.007) | 174.2 (8.84) | 8.9 (0.6) | 0.578 (0.078) | 1.196 (0.025) |
| T3 | 47.72 (1.05) | 0.2304 (0.01) | 106.2 (8.16) | 7.28 (0.74) | 0.74 (0.05) | 1.256 (0.034) |

$\lambda$ - thermal conductivity; $\alpha$ - thermal diffusivity; $\beta$ - thermal absorptivity; $\tau$ - thermal resistance; $h$ - thickness; $q$ max—maximum stationary heat flux density

$f_i$—area factor of section i of the nude manikin [29].

Although the Parallel Model is recommended for standard tests [23], the Serial method was used in this study, as the manikin´s body was not homogeneously covered with the blanket.

This study was approved by the Executive Board and the Ethics Committee of the institution where the study was carried out (CEUOSSCMP/2014).

## Results

Differences were found in the thermal comfort properties of the three samples proposed for the inner layer, as presented in Table 3.

Sample T1 showed the highest thermal conductivity, lowest thermal diffusivity and thermal resistance, highest thickness, and lowest maximum stationary heat flux density. Then, the thermal insulation ability of the three-layered sets (A+B+T1 to A+B+T3) was tested and calculated. The experimental results of the measurements are listed in Table 4.

Results show the lowest heat loss values and the best thermal insulation for Set 1.

One-way analysis of variance was used to determine if there were significant differences in thermal insulation between the three sets. The results are presented in Table 5.

## Discussion

Three textile sets, each composed of three fabrics, were tested to analyse the best thermal insulation performance to create a new blanket for future testing in the perioperative context. The fabrics for the mid and outer layers were previously selected and tested. This study aimed to test the thermal characteristics of the fabrics proposed for the inner layer and investigate how each inner layer fabric influenced thermal insulation performance of the three-layered sets.

Set 1 scored the best thermal insulation results, with its inner layer fabric composed of polypropylene, polyamide, and elastane.

In Set 1, the polypropylene (inner layer) is in direct contact with the skin, thus providing better sensorial comfort due to the softness and excellent moisture control properties of this fibre. Polypropylene is one of the major synthetic fibres and more resistant to attacks from insects, moths, moulds, fungi, and many everyday chemicals. In addition, polypropylene is dyed during extrusion, thus avoiding the loss of these chemicals to the patient's skin during its

Table 4. Heat loss (P), thermal resistance (R) and thermal insulation (Clo) of three-layered sets.

| | P (W/m²) (M/SD) | | R (m²/kw) | Clo |
|---|---|---|---|---|
| Type | Group A | Group B | Serial | Serial |
| Set 1 (A+B+T1) | 61.889 (5.899) | 22.344 (2.146) | 0.546 | 3.522 |
| Set 2 (A+B+T2) | 63.981 (4.022) | 24.245 (2.633) | 0.445 | 2.935 |
| Set 3 (A+B+T3) | 79.476 (3.187) | 30.341 (1.401) | 0.384 | 2.477 |

**Table 5. ANOVA for thermal insulation values.**

|  | DF | SS | MS | F | *p* |
|---|---|---|---|---|---|
| Between groups | 2 | 0.016 | 0.008 | 44.694 | <0.001 |
| Intragroup | 57 | 0.010 | 0.000 |  |  |
| Total | 58 | 0.026 |  |  |  |

Statistically significant differences were found between sets. LSD Post Hoc tests showed differences between sets 1 and 2, between sets 1 and 3, and between sets 2 and 3.

use, and it is hypoallergenic. Knitting polypropylene with polyamide enhances the durability and strength of the fabric, as polyamide has excellent elasticity and similar healthy properties as polypropylene. These factors combined are relevant for the objective of this practical solution. In addition, polyamide promotes a good thermal comfort sensation. Also, elastane improves elasticity contributing to increased comfort levels [30–32].

The fabric—inner layer sample T1—scored the highest for thermal conductivity. This means that this fabric conducts more heat through its surface than the others, allowing the heat emitted by the person to pass faster through its textile fibres [26]. This is an important feature of the inner layer because this layer is intended to allow the heat to go through the system and remain inside. The high air permeability of the mid-layer allows air to circulate within the system. The air permeability close to zero of the outer layer preserves the air inside of the system and prevents air from escaping to the outside. The whole system maintains the person warm and promotes a good thermal sensation.

Sample T1 also showed the highest thickness values. Evidence reports a relationship between the value of the thermal resistance of materials and their thicknesses [26, 33]. The reason thickness increases thermal resistance is that the air layer inside the thermal barrier fabrics is higher because of its bulky and thick structure, resulting in heat transfer-resistant layers [26, 33].

Thermal absorptivity and maximum heat flux provide the expected sensation to touch, whereas a warm feeling is obtained when these two variables are lower [34–36]. Sample T1 had the lowest maximum heat flux and the second lowest thermal absorptivity, whereas sample T3 presented the lowest value. Nevertheless, the behaviour of the three materials in each set was fundamental for designing the final product. The black colour of the inner layer tested in Set 1 corroborates that when dark in colour, the inner layer absorbs more heat, keeps the body warmer, and dries faster [20].

This study revealed changes in the thermal resistance of the three layers compared with thermal resistance of the separate inner layers. Previous studies on the measurement of single textile materials have revealed that it is possible to predict thermal resistance and equivalent thermal conductivity of systems with two layers [26].

Because both the mid and outer layers have been the same in all sets, whereas the inner layer was different in each set, these results suggest that the inner layer may be a determining factor for increasing thermal insulation of the whole set.

This study sought to combine existing fabrics to design a new blanket rather than developing new fabrics or materials. This innovative approach takes advantage of different fabrics and features: a three-layered blanket instead of three garments. However, this study is about a new passive measure, a three-layered blanket which is a new fabric combination that does not exist in clinical practice yet. Therefore, this solution needs to be clinically tested in a perioperative context. Notably, no evidence was found of a textile thermal insulation system to protect patients in the perioperative context with these characteristics.

The blanket solution is set to be tested. If it produces positive results in the clinical setting, a new type of thermal protection for patients in the perioperative context will be made available.

Notwithstanding the innovative findings, this study has limitations, such as the need to calculate the costs related to developing the blanket solution, its use, and the durability and resistance of the fabrics to washing to assess the product's cost-effectiveness.

## Conclusion

The thermal comfort properties of three-layered sets composed of the same mid and outer layers and a different inner layer were tested to find the best thermal comfort performance using a thermal manikin.

The set whose inner layer was composed of polypropylene, polyamide and elastane revealed the best thermal insulation levels, which indicates that the set is more suitable and provides more effective thermal protection compared with the other tested sets.

In view of these findings, the blanket will be tested with patients in the perioperative context, comparing its effectiveness for thermal comfort with the forced warm air system recommended by international guidelines.

## Author Contributions

**Conceptualization:** Isaura Carvalho, Miguel Carvalho.

**Data curation:** Isaura Carvalho.

**Formal analysis:** Teresa Martins.

**Methodology:** Isaura Carvalho, Miguel Carvalho.

**Supervision:** Fernando Abelha.

**Validation:** Liliana Fontes.

**Writing – original draft:** Isaura Carvalho.

**Writing – review & editing:** Miguel Carvalho, Teresa Martins, Fernando Abelha.

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
