## [Decision Letter · Decision Letter 0]

18 Apr 2022

PONE-D-21-32944

Perioperative thermal insulation: testing comfort properties for different textile sets

PLOS ONE

Dear Dr. Teresa Martins,

Thank you for submitting your manuscript to PLOS ONE. After careful consideration, we feel that it has merit but does not fully meet PLOS ONE’s publication criteria as it currently stands. Therefore, we invite you to submit a revised version of the manuscript that addresses the points raised during the review process.

**Editor Comments to the Author**

I am returning your manuscript with two reviews. The reviewers came to different conclusions about the paper, as you will see. After reading the reviews and looking at the manuscript, I am afraid that I have to concur with the critical reviews as provided by the reviewers.

Specifically, I have a few major observations on the overall presentation of the work in the manuscript including the lack of clarity in demonstrating the need of the study, parametric specifications and their significance for the system under consideration, the method of evaluation with adequate justification and the data available in the manuscript are not fully convincing.

I am sorry I cannot be more positive at the moment. However, as I have noted, all is not lost. It requires a lot of work and a major revision that I believe you need more time to work on the manuscript for a resubmission if you wish to do so. Please note that it will have to go through the second round of review. Hence, pay attention to the abovementioned suggestions and the following reviewer suggestions and give them due consideration.

You must provide all data underlying the findings in your manuscript fully available for the review process without any restrictions. The data should be provided as part of the manuscript or its supporting information, or deposited to a public repository. If there are restrictions on publicly sharing data—e.g. participant privacy or use of data from a third party—those must be specified. Otherwise your paper cannot be accepted.

We encourage you to submit your revised manuscript by May 26 2022 11:59PM. If you will need more time than this to complete your revisions, please reply to this message or contact the journal office at plosone@plos.org. Please include the following items when submitting your revised manuscript:

We look forward to receiving your revised manuscript.

With Best Regards,

Rajagopalan Parameshwaran, Ph.D., M.E.,

Academic Editor

PLOS ONE

“This work is financed by FEDER funds through the Competitive Factors Operational Program (COMPETE) POCI-01-0145-FEDER-007136 and by national funds through FCT - Portuguese Foundation for Science and Technology under the project UID/CTM/000264 and BD SFRH/BD/79762/2011.

This article was supported by National Funds through FCT - Fundação para a Ciência e a Tecnologia, I.P., within CINTESIS, R&D Unit (reference UIDB/4255/2020).”

“TM

Grant number: UIDB/4255/2020).

FCT - Portuguese Foundation for Science and Technology under the project UID/CTM/000264 and BD SFRH/BD/79762/2011. https://www.fct.pt/

The funder did not play any role in the study design.”

**Comments to the Author**

Reviewer #1: Dear authors

The study "Perioperative thermal insulation: testing comfort properties for different textile sets" investigated the thermal properties of inner layer fabrics to be used as garments for patients during surgery. Patients seem to suffer from low temperatures in operation theatres. However, no detailed evidence was provided in the manuscript about this issue. Furthermore, it was unclear how the study should provide new insights for developing novel multi-layer fabrics for perioperative use. Some fabrics were selected as inner layers (without a rationale for the selection). These fabrics were combined with an inter-mediate and an outer layer (again, without further information if these are standard fabrics used in this setting). Benchmark tests were applied on the single layers, which differed between inner layer fabrics and intermediate/outer layer fabric (a thermal characterization of the inner layers without providing de-tailed information about the methodology applied and the calculation of the parameters; air permeability for intermediate/outer layer). Unfortunately, there was no explanation for why different tests were chosen for different layers. In addition, thermal manikin tests were conducted to test multi-layer systems. Two different approaches were selected for the calculation of thermal resistance. Unfortunately, no information was provided on why both methods were considered and what kind of information can be extracted from one or the other approach. Data from benchmark tests and manikin were only compared descriptively (no statistical analysis was applied to the data). Finally, the conclusion was drawn that using a three-layers set may be more suitable and provide more effective thermal protection than the other tested fabrics in patients during the perioperative period. This conclusion is not based on the experimental results from this study, as the effect of the inner layer fabric on the three-layered fabric system was investigated. No information or data was provided about other fabrics/garments used for patients during surgery.

This manuscript lacks a clear structure and explanation of the study rationale and the goals. The mate-rials and methods section has to provide more detailed information about the calculation of the thermal properties of fabrics and the rationale for selecting fabrics. Furthermore, there is no statistical analysis of data measured in this study. The discussion section is weak and does not consider findings from other studies but presents data from this study superficially (only one study cited in this section).

For the reasons mentioned above, this manuscript does not fulfil the required scientific quality to be published in PLOS ONE.

Reviewer #2: Congratulations for the relevant paper which adresses a very important topic.

The paper is well written.

Regarding the Results section, why do you state that "Although the tests of the inner layer were focused on its thermal conductivity, all its tested thermal properties were included as additional information in this study. However, the

discussion will focus on thermal conductivity only"? (line 200). Why focus on thermal conductivity only?

I would suggest that a statement about the costs of the fabrics should be added to your discussion.

During the discussion, I would suggest that you explicit the main limitations of your study and how this study can translate to clinical practice.

---

## [Author Response · Author response to Decision Letter 0]

18 Oct 2022

We would like to thank you and the Reviewers for all the comments and suggestions on the manuscript and for the opportunity to review and resubmit this work. The comments and suggestions have enhanced the quality of this manuscript. 

we are grateful for all contributions given by the reviewers. 

Editor response:

All the manuscript was reviewed, and new background information was inserted in the Introduction, Method, Results and Discussion sections about reasons for conducting the study, the significance, and parameters of the system under development, the goals of the study and the evaluation method. Data were corrected and explained.

Since the manuscript was reviewed thoroughly and substantial changes were made, they are not copied here.

Response to Reviewer 1:

Information was clarified in the article. 

In the introduction:

1) The study investigated the thermal properties of the inner layer fabrics proposed for a three-layered blanket composed of three fabrics (one layer each). 

2) Evidence was added to show the incidence, dimension, and significance of perioperative hypothermia.

Information was added: “This study sought to combine existing fabrics to design a new blanket rather than developing new fabrics or materials. This innovative approach takes advantage of different fabrics and features: a three-layered blanket instead of three garments.”.

Information was added justifying the selection of the fabrics and providing information about types of fabrics and systems used in the perioperative context. It was clarified that the mid and outer layers had been chosen previously and none of these fabrics are used in the perioperative context. 

“The differences between layered clothing in sports activities and patients in the operating room, led to the following adaptations in structure and shape: (1) Use and combine existing fabrics, since the development of textile materials and fabrics is not an objective of this research; (2) developing a new blanket-type solution made with 3 different fabrics rather than layered garments for patients in the lying position; (3) a soft touch fabric used for the inner layer, a low density and highly porous fabric for the mid layer, and an impervious for air fabric for the outer layer; (4) breathability and perspiration of the outer layer would not be considered as important characteristics of this new solution, since the perspiration produced by the person during a surgery is incipient, contrary to the outer layers used in activities under adverse conditions that need to be breathable due to athletes perspiration; (5) a solution capable of mechanical washing and disinfection; (6) the lower cost of the fabrics for testing; (7) define the ease of penetration of hot air inside, its return to the patient and the reduction of the possibility of heat-escape to the environment were defined as the main characteristics of the solution. “

“Fabrics were selected according to the objectives of each layer. When considering the desired fabric air permeability [23,24] the fabrics for the outer and mid layer had been chosen and previously tested following the NP EN ISO 9237, using Textest FX 3300 Air Permeability Tester equipment with a pressure of 100 Pa and a test surface area of 20 cm2.

For the outer layer, the aim was to find a fabric that allowed air retention inside the blanket, preventing its escape to the environment and keeping the patient comfortable. An existing waterproof and slightly heavy fabric had been chosen, and its air permeability was tested. Results showed that the fabric was impervious to air with 0.3 l/m2/s.

For the mid layer, a fabric with a light, voluminous and low-density structure had been chosen and tested for air permeability, considering the need to create a considerable space that allows the accumulation of a cushion for circulating warm air without adding significant weight or volume. The produced results showed that the fabric for the mid layer was very permeable to air with 500 l/m2/s”

“For the inner layer, a friendly touch soft fabric was selected. Three jersey fabrics were chosen with similar touch sensations..”

Many arrangements were made, and excessive and non-relevant information has been removed. 

Information has been clarified in the previous item.

Information was added in the method section: “The thermal insulation of the tested three-layered sets can be calculated in two ways: by adding the area-weighted local thermal insulation to the different body segments of the manikin - serial method - or by using the heat flow from the manikin´s body - parallel method”

“Serial and Parallel Models were used for this study. Although the Parallel Model is recommended for standard tests, in this study, both methods were used to understand the differences between them since the product being tested covered only half of the segments of the manikin”

Proper statistical data were added. ANOVA analysis was performed, and a new table (Table 5) was added.

A major revision of the Discussion section was performed, and proper information was added.

The conclusion section was rewritten to clarify some aspects and match the goals of the study.

“The thermal comfort properties of three-layered sets were tested to find the best thermal comfort performance, using a thermal manikin.

The set whose inner layer was composed of polypropylene, polyamide and elastane revealed the best thermal insulation levels, which indicates that the set is more suitable and provides more effective thermal protection compared with the other tested sets. 

In view of these findings, the blanket will be tested in patients in the perioperative context, comparing its effectiveness for thermal comfort with the forced warm air system recommended by international guidelines.”

Information was added to this topic.

“Two types of body protection are commonly used in the perioperative context: passive measures or thermal insulation systems, which create a barrier to prevent heat loss from the patient's body to the environment - the most commonly used are cotton blankets; and active measures or warming systems, which warm the patient's body through heat from external sources.”

A major revision of the article was performed.

Changes were made and information was added. 

Since the manuscript was reviewed thoroughly and substantial changes were made, they are not copied here.

New information was added to the discussion section considering the study results and updated cited studies.

Since the manuscript was reviewed thoroughly and substantial changes were made, they are not copied here.

Importantly, this is an innovate idea and no previous studies were found on this specific subject. 

Response to Reviewer 2:

We agree with this comment. In fact, all the parameters were important. 

What was written probably led to some misunderstanding. 

Information was revised and changes were made to this topic. 

Information about the study limitations was added to the manuscript.

However, further studies will be conducted to address the other aspects (including the costs) of this innovative product.

---

## [Decision Letter · Decision Letter 1]

6 Dec 2022

PONE-D-21-32944R1Development of a perioperative thermal insulation system: testing comfort properties for different textile setsPLOS ONE

Dear Dr. Martins,

Thank you for submitting your manuscript to PLOS ONE. After careful consideration, we feel that it has merit but does not fully meet PLOS ONE’s publication criteria as it currently stands. Therefore, we invite you to submit a revised version of the manuscript that addresses the points raised during the review process. 

We look forward to receiving your revised manuscript.

Kind regards,

Rajagopalan Parameshwaran, Ph.D., M.E.,

Academic Editor

PLOS ONE

Journal Requirements:

Reviewers' comments:

Reviewer's Responses to Questions

**Comments to the Author**

1. If the authors have adequately addressed your comments raised in a previous round of review and you feel that this manuscript is now acceptable for publication, you may indicate that here to bypass the “Comments to the Author” section, enter your conflict of interest statement in the “Confidential to Editor” section, and submit your "Accept" recommendation.

Reviewer #1: (No Response)

Reviewer #2: All comments have been addressed

2. Is the manuscript technically sound, and do the data support the conclusions?

Reviewer #1: Partly

Reviewer #2: Yes

3. Has the statistical analysis been performed appropriately and rigorously? 

Reviewer #1: Yes

Reviewer #2: Yes

4. Have the authors made all data underlying the findings in their manuscript fully available?

Reviewer #1: Yes

Reviewer #2: Yes

5. Is the manuscript presented in an intelligible fashion and written in standard English?

Reviewer #1: No

Reviewer #2: Yes

6. Review Comments to the Author

Reviewer #1: Dear authors

Thank you very much for your profound revisions of your manuscript. From my point of view, the manuscript was substantially improved regarding the clarity and the scientific presentation of your findings.

I have some further points to be clarified before the manuscript can be considered for publication. Furthermore, the manuscript still includes typos and punctuation errors to be corrected.

P2L33: I would add blanket here so that it becomes clear what kind of textile you are talking about

P2L34: what is meant by "previous selection"? I would say that you just selected the mid and outer layers to investigate how the inner layer affects the thermal protection of the three-layered blanket? (goal of your study)

P3L67: be more specific -> use blankets rather than system unless there are other structures to be considered. In this case, please mention all of them

P3L71ff: unclear sentence

P4L74: one study demonstrated (not some authors demonstrated)

P4L77: be more precise with your terminology; what is meant by devices (active heating?); what is meant by systems (passive heating such as blankets?)

P4L79: what is the outcome of studies 15 and 16?

P4L80: see comment P4L77

P4L97ff: talk about layers; this section provides correct information about the 3-layered clothing system for outdoor activities. You do not need to provide that many details as this information is irrelevant to your study.

P5L117: what is meant by "three fabrics (one layer each)" -> I think it is clear that a fabric is a single-layered material

P5L119: selected -> please provide some information about why you previously decided on the mid and outer layer; what were the criteria for selection? -> this information might be provided in the methods section

P6L126: this study investigates the effect of the inner layer of a thermal protective blanket on its thermal insulation performance... you do not need to elaborate on why it differs from sports clothing

P6L126f: why is the use and combination of existing fabrics distinguishing between sports clothing and your blanket? Sports clothing could also be made of existing fabrics…

P6L135f: unclear what is meant by "lower cost of the fabrics for testing."

P8L180: Please provide details for the Alambeta apparatus (producer, country)

P9L197: Please provide details for the thermal manikin (producer, country)

P10L213: stability of manikin surface temperature? +/- 0.1°c?

P10L213: range of temperature that the chamber can maintain (+/- 0.5°C?)

P10L214: what do you mean by approximately? Give a range for the relative humidity during the experiment, e.g. 37-47% (as it was monitored continuously)

P10L224f: I would propose using the serial method only, as the manikin's body was not homogeneously covered with the blanket

P12L259ff: is this an aim of your study?

Table 3: Please provide SI-units for your parameters

Table 4: standard deviation for mean values?

P16L335ff: omit this section; this is just a repetition of the paper by Kuklane et al. 2012 with no interpretation of your results

P17L358: however, the 3-layered blanket could further be optimized by considering adjustments in the middle and outer layer?

Reviewer #2: The authors adressed the previous comments sent by the reviewers.

They have made important changes to the manuscript.

7. PLOS authors have the option to publish the peer review history of their article (what does this mean?). If published, this will include your full peer review and any attached files.

Reviewer #1: No

Reviewer #2: No

---

## [Author Response · Author response to Decision Letter 1]

18 Jun 2023

Dear Professor Rajagopalan Parameshwaran,

Academic Editor of PLOS ONE

We are resubmitting the manuscript “Development of a perioperative thermal insulation system: testing comfort properties for different textile sets” for publication in PLOS ONE, according to instructions. We would like to thank you and the Reviewers for all the comments and suggestions on the manuscript and for the opportunity to review and resubmit this work. The comments and suggestions have enhanced the quality of this manuscript. This rebuttal letter shows the changes made and the explanation for each item. The changes and comments are highlighted in yellow.

---

## [Decision Letter · Decision Letter 2]

7 Aug 2023

PONE-D-21-32944R2Development of a perioperative thermal insulation system: testing comfort properties for different textile setsPLOS ONE

Dear Dr. Martins,

Thank you for submitting your manuscript to PLOS ONE. After careful consideration, we feel that it has merit but does not fully meet PLOS ONE’s publication criteria as it currently stands. Therefore, we invite you to submit a revised version of the manuscript that addresses the points raised during the review process.

Please give due consideration to the reviewer comments and address them very carefully so as to consider the revised version of your manuscript for possible publication in PLOS ONE. 

We look forward to receiving your revised manuscript.

Kind regards,

Rajagopalan Parameshwaran, Ph.D., M.E.,

Academic Editor

PLOS ONE

Journal Requirements:

Reviewers' comments:

Reviewer's Responses to Questions

**Comments to the Author**

1. If the authors have adequately addressed your comments raised in a previous round of review and you feel that this manuscript is now acceptable for publication, you may indicate that here to bypass the “Comments to the Author” section, enter your conflict of interest statement in the “Confidential to Editor” section, and submit your "Accept" recommendation.

Reviewer #1: (No Response)

2. Is the manuscript technically sound, and do the data support the conclusions?

Reviewer #1: Partly

3. Has the statistical analysis been performed appropriately and rigorously? 

Reviewer #1: Yes

4. Have the authors made all data underlying the findings in their manuscript fully available?

Reviewer #1: Yes

5. Is the manuscript presented in an intelligible fashion and written in standard English?

Reviewer #1: Yes

6. Review Comments to the Author

Reviewer #1: Dear authors

Thank you for considering my input for finalizing the manuscript. Unfortunately, your revisions did not resolve all ambiguities. Please see my comments below regarding this.

P2L35: as you are going to investigate the effect of the inner layer, there should be information provided in the abstract. Furthermore, it would be beneficial to know what kind of 2nd and 3rd layers you selected (e.g. if these are conventional fabrics used in perioperative thermal insulation systems)

P5L102: the expression "three similar fabric samples" is not very scientific. Why are you going to investigate "similar" fabrics? You should be more interested in how differences between those fabrics affects the thermal insulation performance of the 3-layered blanket. What is the rationale you selected these three fabrics to be investigated?

P5L106: unclear to which blanket concept you are referring to -> (passive) thermal insulation systems

P5L112 do you refer to perspiration of the fabrics (e.g. substances released from the fabrics) or moisture management of sweat?

P5L118: it still does not become clear why the blanket concept should be based on "mountain clothing" -> there is no clear information provided in Figure 1

P6L121: the inner layer is introduced in a general manner even though the study is about the investigation of the effect of the inner layer on total thermal protection. I would expect some more detailed in-formation and explanations

P10L203 I'm a bit astonished that you added exactly the temperature ranges I proposed for the manikin and the climatic chamber. I hope you did not just add them to the text, but they represent the accuracy of your devices.

7. PLOS authors have the option to publish the peer review history of their article (what does this mean?). If published, this will include your full peer review and any attached files.

Reviewer #1: No

---

## [Author Response · Author response to Decision Letter 2]

20 Aug 2023

Dear Professor Rajagopalan Parameshwaran,

Academic Editor of PLOS ONE

We are resubmitting the manuscript “Development of a perioperative thermal insulation system: testing comfort properties for different textile sets” for publication in PLOS ONE, according to the reviewers’ instructions. We would like to thank you and the Reviewers for all the comments and suggestions on the manuscript and for the opportunity to review and resubmit this work. The comments and suggestions have enhanced the quality of this manuscript. This rebuttal letter shows the changes made and the explanation for each item. 

Comments to the Author 

Reviewer1 

P2L35: as you are going to investigate the effect of the inner layer, there should be information provided in the abstract. Furthermore, it would be beneficial to know what kind of 2nd and 3rd layers you selected (e.g. if these are conventional fabrics used in perioperative thermal insulation systems) 

The text was corrected, and some information is added to the abstract:

“Three fabric samples for the inner layer with same soft touch and different textile compositions were tested to find its effect on increasing the thermal insulation of the whole set, using a thermal manikin.”

“The best thermal insulation and thermal comfort performance was obtained by the set using an inner layer composed of polypropylene, polyamide, and elastane whose results were the highest thermal conductivity and thickness and the lowest maximum stationary heat flow density.”

“The results indicated that this fabric influenced positively the values of the whole set once increased its thermal protection.”

P5L102: the expression "three similar fabric samples" is not very scientific. Why are you going to investigate "similar" fabrics? You should be more interested in how differences between those fabrics affects the thermal insulation performance of the 3-layered blanket. What is the rationale you selected these three fabrics to be investigated? The text was corrected:

“three fabric samples for the inner layer with same soft tactile sensation and different textile compositions were tested” 

Information regarding the characteristics and criteria for selecting the inner layer fabrics has been provided within the “Material and Methods” section.

P5L106: unclear to which blanket concept you are referring to -> (passive) thermal insulation systems Is clarified in the text that the blanket is a passive measure or a thermal insulation system. They are two different designations to the same concept. 

“passive measures or thermal insulation systems, which create a barrier to prevent heat loss from the patient's body to the environment, such as cotton sheets or cotton blankets…”

“The blanket is a thermal insulation system because it creates a barrier to prevent heat loss from the patient's body to the environment.”

P5L112 do you refer to perspiration of the fabrics (e.g. substances released from the fabrics) or moisture management of sweat? According to suggestion information is added, and it was clarified that the perspiration was to the patient´s sweat.

“although the fabrics used in extreme activities have technical properties such as breathability and perspiration, those characteristics would not be considered as important for this new solution since the humidity of the sweat produced by the person during surgery is incipient”

P5L118: it still does not become clear why the blanket concept should be based on "mountain clothing" -> there is no clear information provided in Figure 1 To build a better passive thermal body protection system for patients in the perioperative context, the authors were inspired by the mountain clothing model. The figure 1 intended to show the scheme of the overlap of the fabrics. Some clarifications were added to the text before and after the Fig. 1 presentation, within the “Material and Methods” section.

P6L121: the inner layer is introduced in a general manner even though the study is about the investigation of the effect of the inner layer on total thermal protection. I would expect some more detailed information and explanations Additional information and explanations regarding the inner layer have been included in the manuscript, within the “Material and Methods” section.

P10L203 I'm a bit astonished that you added exactly the temperature ranges I proposed for the manikin and the climatic chamber. I hope you did not just add them to the text, but they represent the accuracy of your devices. Tests were conducted in an adiabatic chamber under controlled environmental conditions.

Variance: the range of values for the manikin´s temperature is 33ºC (±0.008), which was rounded to 0.01. Unfortunately, 0.1 was mistakenly used instead of 0.01.

The range of values for the relative humidity is 42% (±1.8) which was rounded to 2.

The range of room temperature values is correct: 22ºC (±0.5). 

We would be pleased to provide the databases if You would like to consult them.

 The figure has been changed. The reviewer is quite right to point out that an explanation of it would be necessary. However, we consider that the simpler figure is more adapted.

---

## [Editor Report · Decision Letter 3]

30 Aug 2023

Development of a perioperative thermal insulation system: testing comfort properties for different textile sets

PONE-D-21-32944R3

Dear Dr. Martins,

We’re pleased to inform you that your manuscript has been judged scientifically suitable for publication and will be formally accepted for publication once it meets all outstanding technical requirements.

Kind regards,

Rajagopalan Parameshwaran, Ph.D.

Academic Editor

PLOS ONE
---

## [Editor Report · Acceptance letter]

4 Sep 2023

PONE-D-21-32944R3 

Development of a perioperative thermal insulation system: testing comfort properties for different textile sets 

Dear Dr. Martins:

I'm pleased to inform you that your manuscript has been deemed suitable for publication in PLOS ONE. Congratulations! Your manuscript is now with our production department. 

Kind regards, 

on behalf of

Dr. Rajagopalan Parameshwaran 

Academic Editor

PLOS ONE